# Association of Anxiety Awareness with Risk Factors of Cognitive Decline in MCI

**DOI:** 10.3390/brainsci11020135

**Published:** 2021-01-21

**Authors:** Ariela Gigi, Merav Papirovitz

**Affiliations:** Psychology & Behavioral Science Department, Ariel University, Ariel 44837, Israel; meravpapi@gmail.com

**Keywords:** anosognosia, memory decline, alexithymia, electrodermal activity, objective anxiety measurements

## Abstract

Studies demonstrate that anxiety is a risk factor for cognitive decline. However, there are also study findings regarding anxiety incidence among people with mild cognitive impairment (MCI), which mostly examined general anxiety evaluated by subjective questionnaires. This study aimed to compare subjective and objective anxiety (using autonomic measures) and anxiety as a general tendency and anxiety as a reaction to memory examination. Participants were 50 adults aged 59–82 years who were divided into two groups: MCI group and control group, according to their objective cognitive performance in the Rey Auditory Verbal Learning Test. Objective changes in the anxiety response were measured by skin conductivity in all tests and questionnaires. To evaluate subjective anxiety as a reaction to memory loss, a questionnaire on “state-anxiety” was used immediately after completing memory tests. Our main finding was that although both healthy and memory-impaired participants exhibited elevations in physiological arousal during the memory test, only healthy participants reported an enhanced state anxiety (*p* = 0.025). Our results suggest that people with MCI have impaired awareness of their emotional state.

## 1. Introduction

Anxiety symptoms have been extensively studied among people with cognitive decline [1,2]. Anxiety has been identified as a risk factor for memory loss in mild cognitive impairment (MCI) [3], so understanding the components of anxiety in the disorder is critical. Moreover, subjective memory complaints (SMCs) and seeking help in memory clinics are associated with anxiety, even more than with an objective cognitive condition [4,5,6]. In doing so, the association of anxiety with SMC and help-seeking may be a key element in the early diagnosis of cognitive decline [4].

The notion that anxiety is common among people with cognitive deterioration is widely accepted [1,2,7,8]. However, the prevalence of anxiety among people with MCI is unclear. Studies have found varying anxiety incidence in predementia stages [1]. Rozzini et al. [9] identified symptoms of anxiety with a prevalence of 74% among people diagnosed with MCI, while Geda et al. [10] identified symptoms of anxiety with an incidence of only 11% among people with MCI. The differences in these findings are caused by differences in the sampling methods, the criteria for diagnosing MCI, and tools used to examine anxiety and their level of sensitivity [1,11].

To the best of our knowledge, two aspects of anxiety among MCI participants have received scarce attention in studies. The first aspect is the differentiation between anxiety as a personality feature and anxiety as a reaction to memory deterioration. Although using different questionnaires, studies of anxiety among MCI participants mostly address general anxiety and do not examine whether anxiety elevates as a reaction to memory loss. A meaningful differentiation was observed in a recent paper, in which Gigi et al. [4] compared medical help seekers (HS) from memory clinics and non-help seekers (NHS). Gigi et al. assessed people’s anxiety triggered by facing their memory deterioration (even minor normal decline). To assess this specific anxiety, they used a “state-anxiety” questionnaire (assessing anxiety responses to stressful situations; [12]) immediately after completing memory tests. In addition, a “trait-anxiety” questionnaire was administered (assessing general anxiousness). It was found that HS reported higher levels of state anxiety compared to NHS in contrast to general anxiousness, in which no differences were found.

The second aspect is the distinction between objective anxiety and subjective anxiety. Despite the use of various tools to measure anxiety, most studies have focused on self-reported anxiety, which is subjective anxiety. Yet, there is little reference to the assessment of objective anxiety, which can be indicated by elevated levels of physiological arousal, assessed through physiological measures, such as electrodermal measures (e.g., skin conductance response) [13]. Studies that examined different types of anxiety (e.g., test anxiety) have found that since physiological responses associated with anxiety are difficult to control, physiological measures may represent an objective way of assessing participants’ arousal [13]. Skin conductance is measured by using the galvanic skin response (GSR), where levels of palm sweat are measured in relation to emotional response. It was found that anxiety affects GSR levels [14].

The purpose of this study was to examine these two aforementioned aspects: we aimed to compare subjective anxiety with objective anxiety (autonomic measures) and general anxiety and anxiety as a reaction to memory deficits (even if minor). To assess subjective anxiety as a reaction to memory condition, we used a “state-anxiety” questionnaire immediately after completing memory tests. An indication of objective anxiety was obtained from measuring changes in physiological arousal by using GSR. Assuming that previous findings showing a high incidence of anxiety among MCI people are more reflective of reality, we hypothesized that participants who exhibit cognitive decline would present higher subjective and objective anxiety in comparison to the healthy participants. We hypothesized that all participants would exhibit an increase in state anxiety following the memory test, with no difference between the two groups.

## 2. Materials and Methods

### 2.1. Participants

Fifty older adults (aged 59–82 years) volunteered to participate in this study. Participants were recruited from the community using a snowball method with acquaintances or through elder-community centers (which serve as a social meeting place for healthy older adults from the community).

Participants were divided into two groups according to their objective cognitive performance (see objective memory evaluation section): MCI group (*N* = 9; mean age = 70.8 years; SD = 7.5) and control group (*N* = 41; mean age = 69.2 years; SD = 5.6). Inclusion criteria for both groups were native Hebrew speakers and older than 50 years of age. For both groups, exclusion criteria were a Rey Auditory Verbal Learning Test (RAVLT) score of under 2 SD (to avoid dementia) and any neurological condition or psychiatric diagnosis (including anxiety, depression or taking medication for anxiety or depression) during the last five years (self-reported). For assigning participants to the MCI group, we adhered to the diagnostic criteria for MCI as defined by Petersen and colleagues [15], though we did not require subjective memory impairment (see criteria in Solfrizzi’s paper, [16]). For the MCI group, inclusion criteria were: being lower than one standard deviation on a memory test (adjusted for age and education norms); no difficulties with activities of daily living (ADLs) (according to the experimenter’s unstructured interview) and no dementia.

### 2.2. Neuropsychological Testing

All participants underwent a neuropsychological test battery that began with a demographic questionnaire and was followed by objective and subjective memory and anxiety assessment.

#### 2.2.1. Memory Measurement

The Hebrew version of the RAVLT was used to evaluate the objective abilities of episodic learning and memory [17]. A list of 15 words was displayed auditorily five times (trials), and at the end of each time, participants were requested to report as many words as they remember. Twenty minutes later, they were asked to retrieve the words from memory. Two measures were evaluated by the RAVLT: one was the total learning (sum of Trials 1–5), reflecting the individual’s ability to accumulate words across repeated learning trials; the other was delayed memory (long term memory), as measured after 20 min. Within the Hebrew version of the RAVLT [17], we used delayed memory, recognition and total learning (trials 1–5) measurements to evaluate participants’ cognitive state. These measures were normed for age and education for each participant. The cutoff score for assignment to the MCI group was 1SD (an example for this cutoff in Ganguli’s paper, [18]).

#### 2.2.2. Objective and Subjective Anxiety Measurements

The state-trait anxiety inventory (STAI) includes the trait anxiety questionnaire and state anxiety questionnaire [12]. The trait anxiety questionnaire was designed to evaluate a stable tendency to experience anxiety across various situations. Therefore, it was administered following the demographic questionnaire. The state-anxiety questionnaire, however, evaluates anxiety experienced at that moment [12]. Since the aim of our study was to measure anxiety associated with memory functioning, the state anxiety questionnaire was administered immediately following the memory testing. More precisely, the trait anxiety questionnaire was administered after the demographic questionnaire, and the state anxiety questionnaire was administered immediately after the learning phase (after trial 5).

GSR is an indicator of physiological arousal, measured by a GSR device (Prorelax interactive program, Mindlife, Jerusalem, Israel). The device was found to provide dependable measures of autonomic arousal levels in various contexts [19,20]. Two 5-mm-diameter electrodes were placed on the index and middle finger of the participant’s non-dominant hand. Electrodes were connected to a sensor and a receiver. An isolated skin conductance coupler applied a constant 0.5 V potential across the electrode pair. The finger sensors measured the changes in the skin that are affected by sweat gland activity in response to emotional and mental states [19,20]. Arousal of the sympathetic nervous system produces sweat, and hence, enhances the skin’s capacity to conduct an electric current, and the measured conductance is increased. Arousal data was represented in arbitrary units, as higher GSR scores reflect increased conductivity, indicating greater arousal. The examiner marked in the device the moment the demographic questionnaire and the memory test began and ended. The average of measurements received from the GSR device during the RAVLT and the average of measurements accepted during the demographic questionnaire (representing the baseline level) were analyzed.

### 2.3. Procedure

Community participants were tested individually in a quiet room in their home or a quiet room at their senior center. Clinic participants were tested separately in a quiet room at the memory clinic. All participants filled a demographic questionnaire, a trait anxiety questionnaire, a subjective memory questionnaire ([21]), and an objective assessment of memory ability (the RAVLT). Immediately after the memory test, each participant filled a state anxiety questionnaire. Participants were connected to a GSR device throughout the test and skin conductivity was measured continuously. The experimenter marked through the software the location of the beginning and end of a questionnaire/test. Participants were rewarded with a written report regarding their performance.

### 2.4. Statistical Methods

Due to discrepancies in groups size and since the MCI group was too small to meet criteria for parametric tests, differences between the groups were analyzed using the non-parametric chi-square and Mann–Whitney tests. To examine whether memory test performance raises the level of arousal, we tested the groups for homogeneity of variances (Levene’s test) and performed a mixed-design ANOVA of 2 × 2. Statistical analyses were performed using SPSS (version 25.0, IBM company, Armonk, NY, USA). The level of significance was set at *p* < 0.05.

## 3. Results

The division of the sample into groups was determined according to their total learning scores in the RAVLT test (concerning gender and age norms; [17]). Analyses of demographic characteristics (Table 1) were conducted by using chi-square (for gender) and by Mann–Whitney tests (age and education). No significant differences were observed between the groups in these parameters (*p* >.05). Significant differences were identified between the groups in objective memory performance (RAVLT; z = 4.3; *p* < 0.001;
Table 1).

The results of Leven’s test suggested that the assumption of homogeneity of variances was verified; variances (*p* > 0.05). A 2 × 2 ANOVA was conducted to examine the effects of group (control, MCI; between-subjects) and timing (anxiety assessment while undergoing a memory test and during the demographic questionnaire; within-subjects) on anxiety levels. There was a significant main effect of timing, with anxiety while undergoing the memory test being higher than when filling in the demographic questionnaire (F (1,48) = 36.9; *p* < 0.001; level of arousal during a demographic questionnaire: control 9347, SD: 1444, MCI 9566, SD: 1257; level of arousal during memory test: control 8286, SD: 1363, MCI 8883, SD: 1421). However, there was no main effect for group (F < 1; *p* > 0.05). Thus, no significant difference was observed in the variability of physiological arousal between the groups. In addition, no interaction was observed between factors (F (1,48) = 1.73; *p* > 0.05).

Statistical analysis of memory and affective scores (using Mann–Whitney tests; Table 2) revealed significant differences between groups in state anxiety. The control group reported significantly higher state anxiety levels compared to the MCI group (z = 2.24; *p* = 0.025) and in objective memory performance (RAVLT; z = 4.3; *p* < 0.001). However, the two groups were quite similar in levels of trait anxiety (z = 1.6; *p* > 0.05).

## 4. Discussion

In this study, we examined both subjective and objective anxiety. Comparing the level of physiological arousal (measured by GSR) while undergoing a memory test and the level of arousal measured while completing the demographic questionnaire revealed that both groups experienced increased activity of the sympathetic system during the memory test. However, comparing the results of the state anxiety questionnaires completed immediately after the memory test, we found that only the healthy participants reported significantly higher levels of anxiety. An inconsistency between objective and subjective measurements among MCI participants is evident in the literature concerning memory performance; while there were significant objective differences between healthy and MCI participants in memory abilities, no differences were obtained in SMC questionnaires. A study by Jungwirth et al. [22] revealed that approximately 94% of memory-impaired individuals did not complain about memory problems. Roberts et al. [23] suggested that such inconsistency is due to limited awareness of memory ability.

If so, what is the connection between not complaining about memory deficits and not reporting on anxiety? We suggest that just as there is a discrepancy between objective and subjective memory ability due to an awareness impairment, so too there is a discrepancy with anxiety. Although both groups had an increased level of sympathetic arousal during the memory test, only the control group was aware of it. Our results may suggest that impaired awareness of MCI is not only implying their cognitive condition but their emotional state as well. Other aspects of emotional awareness deficits among people with MCI have been underreported in the past. Yurvun and Smirni [24,25] found significantly more alexithymia (difficulty in identifying, describing and experiencing emotions; [26]) among MCI participants compared to healthy individuals. Moreover, our findings may also explain the varying incidence of anxiety found in predementia stages [1]. The similar scores in the trait anxiety questionnaire vs. the significant differences in the state anxiety questionnaire between the two groups can suggest that different types of anxiety questionnaires, or even the location of the questionnaire in the test battery, might influence the findings.

### Research Implications

Previous studies found that help-seeking in memory clinics is related to an emotional state such as anxiety, not an objective cognitive state [4]. Jorm et al. [27] found that people who seek help in memory clinics have more symptoms of general anxiety than those who do not seek help. Gigi et al. [4] found that while no difference was observed between HS and NHS in cognitive ability, HS exhibited higher levels of state anxiety following the memory test. These findings indicate that several MCI participants are, to some extent, unaware of their emotional condition.

If limited awareness of the emotional condition characterizes people with cognitive deterioration, there is a probability that they will not approach memory clinics and consequently will not be diagnosed at the early stages of the disease. Early diagnosis is significant not only in reducing the costs of healthcare systems but also in effectively assisting in slowing down the progress of illness [28]. Therefore, we recommend raising awareness among primary care doctors regarding the complexity of the situation and the need to refer patients to memory clinics even if they are not worried or anxious.

In contrast, our findings indicate that some people who do not suffer a severe impairment in memory ability still experience elevated levels of anxiety. It is well known that anxiety can be a risk factor for cognitive decline [3], as Petkus et al. [29] even suggest the possibility of anxiety treatment to decrease the risk of developing dementia. Accordingly, further research is needed to examine the effect of antidepressant medications on healthy adults with anxiety symptoms.

## 5. Study Limitations and Future Research

It should be noted that since the MCI group included only a small number of participants, there was no reference to the different levels of cognitive decline in this group. Previous research found that the more cognitive decline there is, the greater the impairment of cognitive awareness [30]. Therefore, it is important to examine and characterize the connection between the severity of cognitive decline and the severity of the impairment in the awareness of the emotional state. Hence, we suggest repeating this study with a larger number of MCI participants and examine this notion. Another limitation that should be acknowledged is the lack of neuropsychological testing and objective medical data, including medications that could have given us a more general and comprehensive view of the participants. Finally, although the participants’ IADLs (instrumental activities of daily living) abilities were assessed, an unstructured interview was used for this purpose. Future studies should include a more careful examination of these abilities (e.g., using a structured questionnaire).

## Figures and Tables

**Table 1 brainsci-11-00135-t001:** Demographic characteristics of groups.

	Gender (F/M)	Age (SD)	Education (SD)	RAVLT, TOL (SD)
Control	10/31	69.2 (5.6)	13.9 (2.9)	49 (6.5)
MCI	3/6	70.8 (7.5)	12.5 (2.4)	34 (5.5)

SD, standard deviation; F/M, female/male; RAVLT, Rey Auditory Verbal Learning Test; TOL, total learning. Analyses of age and education were conducted using Mann–Whitney tests, and analyses of gender were conducted by using chi-square.

**Table 2 brainsci-11-00135-t002:** Group differences in memory and affect assessments.

	Control	MCI	*p*-Value
Trait Anxiety (SD)	31.7 (6.6)	27 (7.3)	N.S
State Anxiety (SD)	40.5 (12.3)	31.4 (5.9)	0.025

N.S: Not significant. Group averages and the standard deviation in brackets. Analyses were conducted using Mann–Whitney tests.

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
