# Peer review of "Association of Anxiety Awareness with Risk Factors of Cognitive Decline in MCI"

_brainsci, 2021, doi:10.3390/brainsci11020135_

Round 1

Reviewer 1 Report

A convenience sample of 50 volunteers around 70 years old (59-82) is divided into a subgroup of MCI and a subgroup with  normal memory functioning based on objective test performance in a verbal learning and memory test. State anxiety and trait anxiety (after memory testing) are assessed with questionnaires. Physiologically, anxiety is assessed with a galvanic skin conductance measure, both during filling in of a general questionnaire (near begin of session)  and during performance of the memory test (near end of session). Authors had as a hypothesis that persons with a MCI profile would show  more anxiety, particularly in the challenging memory test,  but this was not found. Apart from the scores on the objective memory test, the only statistically significant group difference is in state anxiety, found higher in the normal functioning group. Authors suggest that people with MCI have may have impaired awareness of their emotional state.

Comments: The distinction the authors make between anxiety as a response to cognitive deterioration and as a more general characteristic is very relevant. There are several methodological problems with their study which should not let them discard their hypothesis so easily. The step towards suggesting an impaired awareness of emotional state in MCI is premature in my opinion. 

First, there are problems with the sample.  They use a convenience sample of volunteers but from the limited information they give, it is difficult to know what population this sample is supposed to represent. In the Introduction the authors themselves mention that the wide variety in the incidence of anxiety  in MRI (11-74 %) described in the literature,  are caused by differences in sampling methods, criteria for diagnosing MCI. If this is true, and it probably is, the authors should have argued why they have the chosen this specific sampling method and operational definition of MCI.  Based on the latter choice they end up with a very unbalanced group size, and particularly an extremely small MCI group.  What I also miss in the description of the sample is a screening with regard to mood disorders.  Mood may have  a strong effect on the reactivity of the GSR 

Secondly, there are problems with the anxiety measurements.  The GSR appears to be an appropriate tool in principle. But many choices can be made which regard to the exact measures and control conditions.  They use a tonic measure of conductance  in 2 conditions which widely differ in terms of stimulus and response mode and cognitive difficulty.  Also one condition is always in the beginning of the session and the other is near the end.  This implies that no sensible conclusion about differential  effect can be made at all, even if differences between groups had been found.  In my opinion, the only sensible comparisons they could have made are separate non-parametric  tests between the groups during the rest condition and during the memory test. Even then it would be hard to draw any conclusions.  The RAVLT is a very demanding test also for high performing persons because almost no one is able to learn and remember all the words. So, it is possible that high performing adults are stressed as much by their own performance  as low performing subjects.  In real life the situation could be very different, e.g. because the memory of cognitively healthy persons might hardly be taxed by everyday functioning. Even apart from stress, the effects of mental effort and anxiety on psycho-physiological measures can hardly be distinguished. 
A detail point: GSR measures are presented as 5 digit units with a negative sign, implying great precision. On the other hand these units are called arbitrary.  I would suggest the variable to be transformed to  positive values (1- the values used now) and at least  to remove the number after the decimal point ). Also the scores should be considered as a rank order scores because the linearity between anxiety and sweat gland activation can be doubted- so non-parametric statistics must be used here.

Thirdly, it is difficult to interpret the filling in of a self report scale about something which happened  some minutes before in  memory impaired subjects.  For example, it could have been that the cognitively healthy subjects vividly remember the difficulty of the memory test and the effort  they had to spend while in the MCI subjects this memory had already faded.  Similar reservations should also be kept in mind before considering impaired awareness of own emotional state as a separate entity in MCI. 

So concluding, the ideas behind the study are fine but the approach taken and conclusions drawn are too superficial in my opinion.  I suggest the authors use this pilot study as a  step in coming to a study with sufficient methodological control.  

Author Response

Dear reviewer,

Thank you for giving us the opportunity to submit a revised version of our manuscript titled: ‘Is it anxiety or awareness of anxiety? Which is really a risk factor of cognitive decline in MCI?’ to Brain Sciences.

We appreciate the time and effort that have dedicated to providing us a valuable feedback on our manuscript. We are grateful to the reviewers for their insightful comments on our manuscript. We have been able to incorporate changes to reflect most of the suggestions provided by the reviewers. We have highlighted the changes within the manuscript.

Here is a point-by-point response to the reviewers’ comments and concerns.

A convenience sample of 50 volunteers around 70 years old (59-82) is divided into a subgroup of MCI and a subgroup after memory testing) are assessed with questionnaires. Physiologically, anxiety is assessed with a galvanic skin conductance measure, both during filling in of a general questionnaire (near begin of session) and during performance of the memory test (near end of session). Authors had as a hypothesis that persons with a MCI profile would show more anxiety, particularly in the challenging memory test, but this was with normal memory functioning based on objective test performance in a verbal learning and memory test. State anxiety and trait anxiety (not found. Apart from the scores on the objective memory test, the only statistically significant group difference is in state anxiety, found higher in the normal functioning group. Authors suggest that people with MCI have may have impaired awareness of their emotional state.

1Q:

First, there are problems with the sample.  They use a convenience sample of volunteers but from the limited information they give, it is difficult to know what population this sample is supposed to represent. In the Introduction the authors themselves mention that the wide variety in the incidence of anxiety in MRI (11-74 %) described in the literature, are caused by differences in sampling methods, criteria for diagnosing MCI. If this is true, and it probably is, the authors should have argued why they have the chosen this specific sampling method and operational definition of MCI.  Based on the latter choice they end up with a very unbalanced group size, and particularly an extremely small MCI group.

1A: We appreciate your comment and question that has given us an opportunity to clarify an issue that may not have been clear enough. First, you have raised a question about what population this sample is supposed to represent. As we mention in line 72: “Participants were recruited from the community using a snowball method with acquaintances or through elder-community centers.” More precisely, Participants were recruited from the community using a snowball method with acquaintances or through elder-community centers (which serve as a social meeting place for healthy older adults from the community). Thus, we believe that the sample may give a representative reflection of the adults in the general community. We added the relevant information in the text (line 72):

Participants were recruited from the community using a snowball method with acquaintances or through elder-community centers (which serve as a social meeting place for healthy older adults from the community)

Second, for assigning participants to the MCI group, we used Petersen’s accepted criteria for MCI diagnosis. It was important for us to sample people from the general population to avoid the effect of intervening variables that may influence anxiety levels (e.g., although there are no differences in memory abilities between people who seek help in memory clinics and people who don’t seek help, help seekers report of having higher levels of state anxiety which may be directly related to their cognitive condition; [1]).  

Regarding the sample size: We assigned every participant who scored lower than 1SD in memory test to the MCI group. The percentage of people in the population who score less than 1SD is 16%. Hence, out of the 50 people we sampled, nine were assigned to the MCI group. We have changed the participants paragraph to include the criteria by which people were assigned to the MCI group (line 77):

Participants were divided into two groups according to their objective cognitive performance (see objective memory evaluation section): MCI group (N = 9; mean age = 70.8 years; SD = 7.5) and control group (N = 41; mean age = 69.2 years; SD = 5.6). Inclusion criteria for both groups were native Hebrew speakers and older than 50 years of age. For both groups, exclusion criteria were a Rey Auditory Verbal Learning Test (RAVLT) score of under 2 SD (to avoid dementia) and any neurological condition or psychiatric diagnosis during the last five years (self-reported). For assigning participant to the MCI group, we adhered to the diagnostic criteria for MCI as defined by Petersen and colleagues [2], though we did not require subjective memory impairment (see criteria in Solfrizzi’s paper, [3]). For the MCI group, inclusion criteria were: being lower than one standard deviation on memory test (adjusted for age and education norms); no difficulties with activities of daily living (ADLs) (according to the experimenter's unstructured interview) no dementia.

We also added clarification in the 'objective memory evaluation' section and believe it is clearer now. (line 93):

Within the Hebrew version of the RAVLT [15], we used delayed memory and total learning (trials 1-5) measurement to evaluate participants’ cognitive state, through using compatible age and education norms for each participant. The cutoff score for assignment to the MCI group was 1SD (an example for this cutoff in Ganguli’s paper [4]).

2Q:

What I also miss in the description of the sample is a screening with regard to mood disorders.  Mood may have a strong effect on the reactivity of the GSR

2A: You are absolutely right, and we thank you for drawing our attention, as you have given us an opportunity to add clarification to the text. In the manuscript, the information appears as "participants were asked whether they had a neurological or psychiatric diagnosis." In fact, there was a detailed questionnaire in which participants were asked, among other things, about taking any medication. We excluded anyone who reported having a psychiatric diagnosis or taking psychiatric medication. We tried to reduce the information in the text to shorten it. However, following your comment, we now understand that this can be problematic. Thus, we have added this information to the manuscript (line 81):

any neurological condition or psychiatric diagnosis (including anxiety, depression, or taking medication for anxiety or depression) during the last five years (self-reported).

At the same time, it is worth noting that since a trait anxiety questionnaire was administered and none of the participants received a score higher than 2SD from the rest of the group (as assessed by inspection of a boxplot), it is unlikely that the sample contains anyone with an anxiety disorder. 

3Q:

Secondly, there are problems with the anxiety measurements.  The GSR appears to be an appropriate tool in principle. But many choices can be made which regard to the exact measures and control conditions.  They use a tonic measure of conductance in 2 conditions which widely differ in terms of stimulus and response mode and cognitive difficulty. Also one condition is always in the beginning of the session and the other is near the end.  This implies that no sensible conclusion about differential effect can be made at all, even if differences between groups had been found.  In my opinion, the only sensible comparisons they could have made are separate non-parametric tests between the groups during the rest condition and during the memory test. Even then it would be hard to draw any conclusions.

3A: You have raised an important point here. As you mentioned, we assessed arousal in two very different conditions (difficulty-wise, timing-wise, etc.), and hence, the significant difference in arousal level is not surprising. In fact, it is even expected. However, our main conclusion is based on the finding that state anxiety did differ between the two groups.  In addition, we maintained a constant order of tests (no counterbalance), since providing a memory test at the beginning of the session may increase the level of arousal, in a way that may remain so for a long time and unevenly between the various participants. Such a condition could have affected the level of arousal in the other tests. Hence, it was critical to first assess the baseline arousal during the demographic questionnaire. We need to emphasize that this study is a pilot study and, as such, the conclusions drawn in it are preliminary ideas that, of course, require in-depth examination.

4Q:

The RAVLT is a very demanding test also for high performing persons because almost no one is able to learn and remember all the words. So, it is possible that high performing adults are stressed as much by their own performance as low performing subjects.  In real life the situation could be very different, e.g. because the memory of cognitively healthy persons might hardly be taxed by everyday functioning. Even apart from stress, the effects of mental effort and anxiety on psycho-physiological measures can hardly be distinguished.

4A: Thank you for pointing out this important question, which allowed us to think about it again. The RAVLT is indeed a demanding task, and since studies have found that healthy adults are also concerned about their memory abilities, we hypothesized (lines 68-69) that for both groups there would be an increase in state anxiety level following this memory task, which was indeed what we found. Our main conclusion relied on the gap we found between the elevated physiological arousal (in GSR) found in both groups, contrary to a significant disparity in reporting an anxiety experience that was elevated only among the healthy participants. Of course, this conclusion, made under laboratory conditions, cannot be directly applied to any situation dealing with memory loss in daily life. However, this finding is of great significance. It was found that anxiety from cognitive decline affects the seeking of medical help [1], and so we wanted to suggest that people with cognitive decline may not be sufficiently aware of their anxiety, and therefore may not seek help.

5Q:

A detail point: GSR measures are presented as 5 digit units with a negative sign, implying great precision. On the other hand these units are called arbitrary.  I would suggest the variable to be transformed to positive values (1- the values used now) and at least to remove the number after the decimal point).

5A: Thank you for this suggestion. We converted the scores to positive values and removed the number after the decimal point.

6Q:

Also, the scores should be considered as a rank order scores because the linearity between anxiety and sweat gland activation can be doubted- so non-parametric statistics must be used here.

6A: Indeed, GSR measures the changes in the sweat glands, which are an indirect indication of anxiety levels. At the same time, there are many examples of researchers using this measurement while making statistical comparisons in a continuous variable (e.g. [5]).   

7Q:

Thirdly, it is difficult to interpret the filling in of a self-report scale about something which happened some minutes before in memory impaired subjects.  For example, it could have been that the cognitively healthy subjects vividly remember the difficulty of the memory test and the effort they had to spend while in the MCI subjects this memory had already faded.  Similar reservations should also be kept in mind before considering impaired awareness of own emotional state as a separate entity in MCI.

7A: You are absolutely right, and we have, accordingly, changed the paragraph to make this clearer. The trait anxiety questionnaire was administered after the demographic questionnaire, while the state anxiety questionnaire was administered immediately after the learning phase (after trial 5). The methodological consideration was that if the state anxiety questionnaire was administered after the delayed memory, the participants might have forgotten the difficulty they experienced during the test and, therefore, the sensation of anxiety might have faded. However, immediately after the fifth trial, the participant could still remember that he had difficulty with the task and that he had had a problem, and therefore, it was essential to place him at this stage. The point of time at which the questionnaires were administered was added in the manuscript (line 101):

Since the aim of our study was to measure anxiety associated with memory functioning, the state anxiety questionnaire was administered immediately following the memory testing. More precisely, the trait anxiety questionnaire was administered after the demographic questionnaire, and the state anxiety questionnaire was administered immediately after the learning phase (after trial 5).        

8Q:

So concluding, the ideas behind the study are fine but the approach taken and conclusions drawn are too superficial in my opinion

8A: Thank you for your valuable comments. This study is a pilot study, and as such, the conclusions drawn in it are preliminary ideas that require in-depth examination. We feel that both the reference to different types of anxiety and the reference to awareness of emotional state are issues that have not received enough attention so far in the literature, and the purpose of this manuscript is to highlight them, bringing them to the fore.

Of course, the follow-up research that will be carried out will be more comprehensive and will incorporate deeper and broader methodological control.

  1. Gigi, A.; Papirovitz, M.; Vakil, E.; Treves, T. Medical Help-Seekers with Anxiety from Deterioration in Memory are Characterized with Risk Factors for Cognitive Decline. Clin. Gerontol. 2018, 43, 204–208, doi:10.1080/07317115.2018.1527423.
  2. Petersen, R.C.; Smith, G.E.; Waring, S.C.; Ivnik, R.J.; Tangalos, E.G.; Kokmen, E. Mild cognitive impairment: clinical characterization and outcome. Arch. Neurol. 1999, 56, 303–8, doi:10.1001/archneur.56.3.303.
  3. Solfrizzi, V.; Panza, F. Coffee Consumption Habits and the Risk of Mild Cognitive Impairment: The Italian Longitudinal Study on Aging Antidementia Drugs and Factors Associated with All-cause Mortality in Community-dwelling Frail Older Patients with Dementia View project. 2015, doi:10.3233/JAD-150333.
  4. Ganguli, M.; Dodge, H.H.; Shen, C.; DeKosky, S.T. Mild cognitive impairment, amnestic type: An epidemiologic study. Neurology 2004, 63, 115–121, doi:10.1212/01.WNL.0000132523.27540.81.
  5. Shapiro, M.; Melmed, R.N.; Sgan-Cohen, H.D.; Eli, I.; Parush, S. Behavioural and physiological effect of dental environment sensory adaptation on children’s dental anxiety. Eur. J. Oral Sci. 2007, 115, 479–483, doi:10.1111/j.1600-0722.2007.00490.x.

Reviewer 2 Report

There are major issues with the methodology that should be addressed. 

The largest problem is that the authors did not identify a cited consensus for a diagnosis of MCI (e.g., Peterson criteria or others). The manuscript states "Participants were divided into two groups according to their objective cognitive performance (see objective memory evaluation section);" however, unless a protocol for MCI diagnosis was provided in another paper (which in that case should be explicitly stated), there is no mention of how a consensus diagnosis of MCI came to be. As such, at this time the authors have not supported that their participants are in fact MCI.

Also, the authors did not provide cutoff scores on the RAVLT, and it was unclear whether or not they used just delayed recall or trials 1-5 to diagnosis MCI. While they acknowledge that they generated both of these measures, it says in lines 93-94 that delayed memory measurement was used to assess cognitive states.

The authors should also clarify why recognition was not tallied. One cannot make the assumption that an individual has memory problems in the absence of poor performance on recognition. It is quite common to find low delayed performance and better recognition, which would speak more to a vascular profile than amnesia or memory problems.

Finally, it is necessary to include instrumental activities of daily living in one's assessment of an MCI diagnosis, which is not present in this study. 

Statistical methods:

In order to run an ANOVA with unequal sample sizes, one must first run a test of homogeneity of variance (Levene’s or Bartlett’s test) to assure that variances are equal between treatment group. If they are unequal, the statistical output would be adjusted for in SPSS. This problem (unequal variance) also holds true for Mann-Whitney tests.  Please make sure to add this to your statistical methods, in addition to the other standard assumptions necessary for these analyses.

Other considerations:

The authors did not provide the names of the self-reported memory questionnaires or the analyses used for this questionnaire in their study. 

Lines 153-157: The authors mentioned earlier that they used state anxiety one time, after the memory assessment. However, specifically in lines 155-157, it reads as though they provided the ‘state anxiety’ questionnaire twice (higher anxiety levels compared to…as well as in…). Please clarify.

Please be specific in describing at what time points (e.g., after trial 5 or delayed recall) the anxiety measures were provided. Currently, it says "after memory testing" which is quite vague (is it after trial 5 or after delayed recall?). 

Lines 184-185: "Moreover, similar findings in the assessment of trait anxiety vs. the significant differences in state anxiety between the two groups can explain the varying results in the literature regarding anxiety in MCI participants." Please clarify what this means?

Lines 44-45: Substitute the first author of the paper cited instead of “we.”

Line 48-49: “in which no differences were found” – I’m not quite sure what this means. It sounds like earlier in the sentence there were differences between group. Do you mean to say that despite higher scores, it did not reach statistical significance?

Grammatical errors. Eg., 176; due to an awareness impairment, so too there is a…

Spell out vs in line 183. Also, there should not be 1 sentence in a paragraph.

There are other major limitations to this study that should be acknowledged. First, the participants provided self-reported diagnoses. It is quite possible that someone has a diagnosis of anxiety and simply did not report. In addition, the lack of information on medications (as mentioned as a future study) should be acknowledged as a limitation. Research has shown that depressive symptoms may be an indicator of MCI. Therefore, it is possible that some of these participants are on SSRI, which may moderate their report of anxiety. Finally, the lack of neuropsychological testing (only RAVLT) and no measures of IADLs is a major limitation as well.  

Author Response

Dear Reviewer,

Thank you for giving us the opportunity to submit a revised version of our manuscript titled: ‘Is it anxiety or awareness of anxiety? Which is really a risk factor of cognitive decline in MCI?’ to Brain Sciences.

We appreciate the time and effort that you have dedicated to providing us a valuable feedback on our manuscript. We are grateful to the reviewers for their insightful comments on our manuscript. We have been able to incorporate changes to reflect most of the suggestions provided by the reviewers. We have highlighted the changes within the manuscript.

Here is a point-by-point response to the reviewers’ comments and concerns.

1Q:

The largest problem is that the authors did not identify a cited consensus for a diagnosis of MCI (e.g., Peterson criteria or others). The manuscript states "Participants were divided into two groups according to their objective cognitive performance (see objective memory evaluation section);" however, unless a protocol for MCI diagnosis was provided in another paper (which in that case should be explicitly stated), there is no mention of how a consensus diagnosis of MCI came to be. As such, at this time the authors have not supported that their participants are in fact MCI.

1A:

We appreciate your comment and question that has given us an opportunity to clarify an issue that may not have been clear enough. For assigning participants to the MCI group, we used Petersen’s accepted criteria for MCI diagnosis. It was important for us to sample people from the general population to avoid the effect of intervening variables (e.g., while there are no differences in memory abilities between people who seek help in memory clinics and people who don’t seek help, help seekers report on higher levels of state anxiety which may be directly related to their cognitive condition; [1]). However, about 16% of the general population score lower than 1SD in memory tests (which is a criterion for MCI diagnosis). Therefore - when sampling people from the general population, we inevitably get groups of unequal size. In our case, since we sampled fifty people from the community, we received an MCI group that included 9 participants. We have changed the participants paragraph to include the criteria by which people were assigned to the MCI group (line 77):

Participants were divided into two groups according to their objective cognitive performance (see objective memory evaluation section): MCI group (N = 9; mean age = 70.8 years; SD = 7.5) and control group (N = 41; mean age = 69.2 years; SD = 5.6). Inclusion criteria for both groups were native Hebrew speakers and older than 50 years of age. For both groups, exclusion criteria were a Rey Auditory Verbal Learning Test (RAVLT) score of under 2 SD (to avoid dementia) and any neurological condition or psychiatric diagnosis during the last five years (self-reported). For assigning participants to the MCI group, we adhered to the diagnostic criteria for MCI as defined by Petersen and colleagues [2], though we did not require subjective memory impairment (see criteria in Solfrizzi’s paper, [3]). For the MCI group, inclusion criteria were: being lower than one standard deviation on a memory test (adjusted for age and education norms); no difficulties with activities of daily living (ADLs) (according to the experimenter's unstructured interview) no dementia.

2Q:

Also, the authors did not provide cutoff scores on the RAVLT, and it was unclear whether or not they used just delayed recall or trials 1-5 to diagnosis MCI.

2A:

We absolutely agree with this comment and, hence, added clarification in the 'objective memory evaluation' section and believe it is clearer now. (line 93):

Within the Hebrew version of the RAVLT [15], we used delayed memory and total learning (trials 1-5) measurement to evaluate participants’ cognitive state, through using compatible age and education norms for each participant. The cutoff score for assignment to the MCI group was 1SD (an example for this cutoff in Ganguli’s paper   [4]).

3Q:

While they acknowledge that they generated both of these measures, it says in lines 93-94 that delayed memory measurement was used to assess cognitive states. The authors should also clarify why recognition was not tallied. One cannot make the assumption that an individual has memory problems in the absence of poor performance on recognition. It is quite common to find low delayed performance and better recognition, which would speak more to a vascular profile than amnesia or memory problems.

3A:

You have raised an important point here. We made a decision to use delayed memory and total learning as measures of cognitive decline due to the findings which demonstrate that there is no significant difference between recall and recognition measures when conducting MCI diagnosis and that they are equally good [5]. It was also found that both delayed memory and recognition measures are related to each other and that the notion that recognition contributes to the diagnosis of amnesia is unjustified [6].

4Q:

Finally, it is necessary to include instrumental activities of daily living in one's assessment of an MCI diagnosis, which is not present in this study.

4A:

You're right, we tried to be brief but that sometimes comes at the expense of important information. We now added the relevant information. We addressed this in response to one of the previous comments. See change in line 82:

For the MCI group, inclusion criteria were….no difficulties with activities of daily living (ADLs) (according to the experimenter's unstructured interview)

5Q:

Statistical methods:

In order to run an ANOVA with unequal sample sizes, one must first run a test of homogeneity of variance (Levene’s or Bartlett’s test) to assure that variances are equal between treatment group. If they are unequal, the statistical output would be adjusted for in SPSS. This problem (unequal variance) also holds true for Mann-Whitney tests.  Please make sure to add this to your statistical methods, in addition to the other standard assumptions necessary for these analyses.

5A:

You are absolutely right, and we thank you for drawing our attention. We refined this sentence and added the homogeneity of variance analysis in the statistical analysis section (line 130):

To examine whether memory test performance raises the level of arousal, we tested the groups for homogeneity of variances (Levene’s test) and performed a mixed-design ANOVA of 2 × 2.

And in the results (line 144):

The results of Leven’s test suggested that the assumption of homogeneity of variances was verified; variances (P > 0.05).

6Q:

Other considerations: The authors did not provide the names of the self-reported memory questionnaires or the analyses used for this questionnaire in their study.

6A:

Thank you for pointing this out. We added a relevant reference to the manuscript, with an explanation that no differences were found between groups and that we did not use these results in the current manuscript (line 121):

All participants filled a demographic questionnaire, a trait anxiety questionnaire, a subjective memory questionnaire ([7]; no results reported, irrelevant for this article), and an objective assessment of memory ability (the RAVLT).

It is important for us to note that we mentioned this questionnaire in the manuscript since we wanted to describe the entire test battery, even though no differences in subjective memory were found between the groups. We did not include details on this questionnaire in the method or in the results because we wanted the manuscript to highlight the distinctive anxiety findings. There is no doubt that the lack of differences we found in subjective memory between healthy and MCI groups is interesting and supports the hypothesis that there is impaired awareness of both cognitive and emotional states. However, because this is a pilot study, we attempted to focus solely on the anxiety findings.

7Q:

Lines 153-157: The authors mentioned earlier that they used state anxiety one time, after the memory assessment. However, specifically in lines 155-157, it reads as though they provided the ‘state anxiety’ questionnaire twice (higher anxiety levels compared to…as well as in…). Please clarify.

7A:

Thank you for bringing this matter to our attention. The word ׳state׳ is indeed missing, so it could be what caused the misunderstanding. We added the word in the relevant place (line 156):

significantly higher state anxiety

8Q:

Please be specific in describing at what time points (e.g., after trial 5 or delayed recall) the anxiety measures were provided. Currently, it says "after memory testing" which is quite vague (is it after trial 5 or after delayed recall?). 

8A:

We agree with your suggestion and have incorporated it throughout the manuscript. The trait anxiety questionnaire was administered after the demographic questionnaire, while the state anxiety questionnaire was administered immediately after the learning phase (after trial 5). The methodological consideration was that if the state anxiety questionnaire was administered after the delayed memory the participants might have forgotten the difficulty they experienced during the test and, therefore, the sensation of anxiety might have faded. However, immediately after the fifth trial, the participant could still remember that he had difficulty with the task and that he had had a problem and therefore it was essential to place him at this stage. The point of time at which the questionnaires were administered was added in the manuscript (line 101):

Since the aim of our study was to measure anxiety associated with memory functioning, the state anxiety questionnaire was administered immediately following the memory testing. Accurately, the trait anxiety questionnaire was administered after the demographic questionnaire, and the state anxiety questionnaire was administered immediately after the learning phase (after trial 5).

9Q:

Lines 184-185: "Moreover, similar findings in the assessment of trait anxiety vs. the significant differences in state anxiety between the two groups can explain the varying results in the literature regarding anxiety in MCI participants." Please clarify what this means?

9A:

We appreciate your comment and question that has allowed us to clarify this paragraph that may not have been clear enough. The paragraph refers to the finding mentioned in the introduction, according to which there are various findings regarding the prevalence of anxiety among people with MCI in the literature. The purpose of the paragraph was to indicate that the differences we found in the different anxiety questionnaires may explain these various findings, as it seems that the type of questionnaire and even its location in the test battery, can influence the reported anxiety. Following your comment, we changed the paragraph to be clearer (line 184):

Moreover, similar findings in the assessment of trait anxiety vs. the significant differences in state anxiety between the two groups can explain the varying results in the literature regarding anxiety in MCI participants. our findings may also explain the varying incidence of anxiety found in pre-dementia stages [8]. The similar scores in the trait anxiety questionnaire vs the significant differences in the state anxiety questionnaire between the two groups can suggest that different types of anxiety questionnaires, or even the location of the questionnaire in the test battery, might influence the findings.

10Q:

Lines 44-45: Substitute the first author of the paper cited instead of “we.”

10A:

Thank you for pointing this out. We changed the text accordingly:

A meaningful differentiation was observed in a recent paper, in which we Gigi et at. [1] compared medical help seekers (HS) from memory clinics and non-help seekers (NHS) [4]. We Gigi et al. assessed people’s anxiety triggered by facing their memory deterioration (even minor normal decline). To assess this specific anxiety, we they used a “state-anxiety” questionnaire (assessing anxiety responses to stressful situations; [12]) immediately after completing memory tests. We It was found that HS reported higher levels of state anxiety compare to NHS in contrast to general anxiousness, in which no differences were found

11Q:

Line 48-49: “in which no differences were found” – I’m not quite sure what this means. It sounds like earlier in the sentence there were differences between group. Do you mean to say that despite higher scores, it did not reach statistical significance?

11A:

Thank you for drawing our attention, as you have given us an opportunity to add clarification to the text (line 43):

A meaningful differentiation was observed in a recent paper, in which Gigi et at. [8] compared medical help seekers (HS) from memory clinics and non-help seekers (NHS) [4]. Gigi et al. assessed people’s anxiety triggered by facing their memory deterioration (even minor normal decline). To assess this specific anxiety, they used a “state-anxiety” questionnaire (assessing anxiety responses to stressful situations; [12]) immediately after completing memory tests. In addition, a “trait-anxiety” questionnaire was administered (assessing general anxiousness). It was found that HS reported higher levels of state anxiety compare to NHS in contrast to general anxiousness, in which no differences were found.

12Q:

There are other major limitations to this study that should be acknowledged. First, the participants provided self-reported diagnoses. It is quite possible that someone has a diagnosis of anxiety and simply did not report.

12A:

Indeed, we relied on the participants' reports. However, since a trait anxiety questionnaire was administered and none of the participants received a score higher than 2SD from the rest of the group (as assessed by inspection of a boxplot), it is unlikely that the sample contains anyone with an anxiety disorder.

13Q:

In addition, the lack of information on medications (as mentioned as a future study) should be acknowledged as a limitation. Research has shown that depressive symptoms may be an indicator of MCI. Therefore, it is possible that some of these participants are on SSRI, which may moderate their report of anxiety.

13A:

You have raised an important point here. As written in the manuscript, participants were asked whether they had a neurological or psychiatric diagnosis. In addition, participants were also asked if they were taking medication, and a participant who reported taking medication for depression or anxiety was excluded from the sample. We added this information in the method (line 82):

any neurological condition or psychiatric diagnosis (including anxiety, depression, or taking medication for anxiety or depression) during the last five years (self-reported).

14Q:

Finally, the lack of neuropsychological testing (only RAVLT) and no measures of IADLs is a major limitation as well. 

14A:

We agree that this is a potential limitation of the study. We have added this as a

limitation in line 213:

Another limitation that should be acknowledged is the lack of neuropsychological testing that could have given us a more general and comprehensive view of the participants.

IADL were assessed by the experimenter's unstructured interview, as difficulties in IADL were exclusion criteria. This was added in line 82:

For the MCI group, inclusion criteria were….no difficulties with activities of daily living (ADLs) (according to the experimenter's unstructured interview)

15Q:

Grammatical errors. Eg., 176; due to an awareness impairment, so too there is a…

Spell out vs in line 183. Also, there should not be 1 sentence in a paragraph.

15A:

Thank you for pointing this out. All spelling and grammatical errors pointed out by the reviewers have been corrected (see line 177 and line 184).

Bibliography:

  1. Gigi, A.; Papirovitz, M.; Vakil, E.; Treves, T. Medical Help-Seekers with Anxiety from Deterioration in Memory are Characterized with Risk Factors for Cognitive Decline. Clin. Gerontol. 2018, 43, 204–208, doi:10.1080/07317115.2018.1527423.
  2. Petersen, R.C.; Smith, G.E.; Waring, S.C.; Ivnik, R.J.; Tangalos, E.G.; Kokmen, E. Mild cognitive impairment: clinical characterization and outcome. Arch. Neurol. 1999, 56, 303–8, doi:10.1001/archneur.56.3.303.
  3. Solfrizzi, V.; Panza, F. Coffee Consumption Habits and the Risk of Mild Cognitive Impairment: The Italian Longitudinal Study on Aging Antidementia Drugs and Factors Associated with All-cause Mortality in Community-dwelling Frail Older Patients with Dementia View project. 2015, doi:10.3233/JAD-150333.
  4. Ganguli, M.; Dodge, H.H.; Shen, C.; DeKosky, S.T. Mild cognitive impairment, amnestic type: An epidemiologic study. Neurology 2004, 63, 115–121, doi:10.1212/01.WNL.0000132523.27540.81.
  5. Bennett, I.J.; Golob, E.J.; Parker, E.S.; Starr, A. Memory evaluation in mild cognitive impairment using recall and recognition tests. J. Clin. Exp. Neuropsychol. 2006, 28, 1408–1422, doi:10.1080/13803390500409583.
  6. Haist, F.; Shimamura, A.P.; Squire, L.R. On the Relationship Between Recall and Recognition Memory. J. Exp. Psychol. Learn. Mem. Cogn. 1992, 18, 691–702, doi:10.1037/0278-7393.18.4.691.
  7. Bennett-Levy, J.; Powell, G.E. The subjective memory questionnaire (SMQ). An investigation into the self-reporting of “real-life” memory skills. Br. J. Soc. Clin. Psychol. 1980, 19, 177–188, doi:10.1111/j.2044-8260.1980.tb00946.x.
  8. Ma, L. Depression, Anxiety, and Apathy in Mild Cognitive Impairment: Current Perspectives. Front. Aging Neurosci. 2020, 12.

Round 2

Reviewer 2 Report

Dear Authors,

Thank you for responding to the question and conversations. The responses were thoughtfully stated. However, there are a few points that still need to be addressed.  

For question 3Q, the authors suggest that “We made a decision to use delayed memory and total learning as measures of cognitive decline due to the findings which demonstrate that there is no significant difference between recall and recognition measures when conducting MCI diagnosis and that they are equally good [5]. It was also found that both delayed memory and recognition measures are related to each other and that the notion that recognition contributes to the diagnosis of amnesia is unjustified [6].” Clinically, one would absolutely need recognition to accurately define someone as amnestic; while the authors were able to find a manuscript supporting their decision, it must be emphasized that, clinically, without recognition, one cannot say that someone has amnesia. Again, there are many people who have retrieval problems, who would perform poorly on trials 1-5 and delayed recall, and do much better on recognition, which would be suggestive of a retrieval problem and not amnesia or memory-problem, per se.  If recognition was not given, it should be discussed as a major limitation such that individuals diagnosed with MCI may not have a memory problem.

Also, while the authors included ADLs, they did not discuss IADLs (instrumental activities of daily living), which is also an important distinction for individuals with MCI. If possible, it is important to include these parameters; if not, it should be discussed as a limitation.

With regards to the sample, is there a way to examine whether or not those who enrolled in the study had subjective memory complaints and therefore increasing the chances of having help seekers in the sample? Like the authors mentioned, it is not uncommon to find those who are ‘worried well’ in memory clinics. This might extend to those who were willing to enroll in a research study that would provide information about their cognitive abilities. This is to say, is there a way to know that the sample is not biased and inclusive of those who had subjective memory complaints? This will be important since the authors are purporting that the increase in state anxiety for the control group is a normal response. Like the other reviewer had mentioned, you would expect higher anxiety, since the test itself is taxing.

Response to Q13: While the authors noted that they excluded individuals with psychiatric d/o, there are plenty of medications for other medical problems that affect anxiety (e.g., gabapentin, hydroxyzine to name a few). So, while they may have excluded participants that communicating having such diagnoses, this does not mean that they excluded those who are on other medications that would reduce anxious distress (even if that was not the intended goal of the medication). Since information was gathered through self-report, it should be stated as a limitation that objective data could not be gathered. 

Line 45-46:

Please change “eg at” to “et al”

Line 87-89: There are grammatical errors

For the MCI group, inclusion criteria were: performing one standard deviation below the mean on a memory test (adjusted for age and education norms); no difficulties with activities of daily living (ADLs; according to the experimenter's unstructured interview); and no dementia.

Author Response

Thank you for investing your time in carefully reading our manuscript.

We have thoroughly reviewed the new comments and have considered each one very seriously. We hope that our responses and revisions adequately address the issues that have been raised.

Q1:

For question 3Q, the authors suggest that “We made a decision to use delayed memory and total learning as measures of cognitive decline due to the findings which demonstrate that there is no significant difference between recall and recognition measures when conducting MCI diagnosis and that they are equally good [5]. It was also found that both delayed memory and recognition measures are related to each other and that the notion that recognition contributes to the diagnosis of amnesia is unjustified [6].” Clinically, one would absolutely need recognition to accurately define someone as amnestic; while the authors were able to find a manuscript supporting their decision, it must be emphasized that, clinically, without recognition, one cannot say that someone has amnesia. Again, there are many people who have retrieval problems, who would perform poorly on trials 1-5 and delayed recall, and do much better on recognition, which would be suggestive of a retrieval problem and not amnesia or memory-problem, per se.  If recognition was not given, it should be discussed as a major limitation such that individuals diagnosed with MCI may not have a memory problem. 

A1:

We appreciate your comment, and we would like to clarify this issue. When administering the RAVLT, we assessed all trials, including recognition trial, to ensure amnesia. Though it is more accurate to present all three memory measures (delayed, total learning, and recognition), the recognition score wasn’t presented since we tried to focus the manuscript as much as possible on the anxiety findings. However, in light of your comment, we understand the need to present this measure as well.  Therefore, the recognition is now presented in the manuscript (line 105):

Within the Hebrew version of the RAVLT [17], we used delayed memory, recognition, and total learning (trials 1-5) measurements to evaluate participants’ cognitive state. These measures were normed for age and education for each participant. 

To further clarify this issue, we have included here two tables that clearly indicate the significant differences in all three memory parameters (delayed, total learning, and recognition).  

Q2:

Also, while the authors included ADLs, they did not discuss IADLs (instrumental activities of daily living), which is also an important distinction for individuals with MCI. If possible, it is important to include these parameters; if not, it should be discussed as a limitation.

A2:

Thank you for pointing this out. We added this limitation to the manuscript (line 251):

Finally, although the participants’ IADLs (instrumental activities of daily living) abilities were assessed, an unstructured interview was used for this purpose. Future studies should include a more careful examination of these abilities (e.g., using a structured questionnaire).

Q3:

With regards to the sample, is there a way to examine whether or not those who enrolled in the study had subjective memory complaints and therefore increasing the chances of having help seekers in the sample? Like the authors mentioned, it is not uncommon to find those who are ‘worried well’ in memory clinics. This might extend to those who were willing to enroll in a research study that would provide information about their cognitive abilities. This is to say, is there a way to know that the sample is not biased and inclusive of those who had subjective memory complaints? This will be important since the authors are purporting that the increase in state anxiety for the control group is a normal response. Like the other reviewer had mentioned, you would expect higher anxiety, since the test itself is taxing.

A3:

We agree that the reviewer has raised an important point here. Although not mentioned it in the text, participants were asked (as part of the demographic questionnaire) whether they have sought medical help for memory decline. 40% of the sample reported that they did seek medical help in the past, with no significant difference between MCI and control groups.

Regarding the notion that the sample was biased and inclusive of those who had subjective memory complaints, as mentioned in line 131, the test battery included a subjective memory questionnaire (SMQ, in which a higher score indicates more complaints). The SMQ scores were not reported in the MS since they were not relevant to the study. However, we would like to point out that the mean score in the SMQ was 138 (SD= 22) with no significant difference between the two groups. In comparison, earlier study that used SMQ in the general population (age ranging between  16-70) reported a mean SMQ of 145.98 (SD=23.43) (as the older participants in that study scored higher than the younger participants) [1].

Considering that the mean score for the sample in our study is lower than the mean score for normal population, this may indicate that our sample is unbiased and includes mainly participants with memory complaints.

Q4:

Response to Q13: While the authors noted that they excluded individuals with psychiatric d/o, there are plenty of medications for other medical problems that affect anxiety (e.g., gabapentin, hydroxyzine to name a few). So, while they may have excluded participants that communicating having such diagnoses, this does not mean that they excluded those who are on other medications that would reduce anxious distress (even if that was not the intended goal of the medication). Since information was gathered through self-report, it should be stated as a limitation that objective data could not be gathered. 

A4:

We agree that this is a potential limitation of the study. We have now added this as a

limitation in line 250:

An additional limitation of the current study is the lack of neuropsychological testing and objective medical data, including medication, that could have given us a more general and comprehensive view of the participants

Q5:

Line 45-46:

Please change “eg at” to “et al”

Line 87-89: There are grammatical errors 

For the MCI group, inclusion criteria were: performing one standard deviation below the mean on a memory test (adjusted for age and education norms); no difficulties with activities of daily living (ADLs; according to the experimenter's unstructured interview); and no dementia.

A5:

Thank you for pointing this out. The spelling and grammatical errors pointed out have been corrected (see line 89 and lines 44-45). In addition, we have carefully proofed the entire MS and corrected all errors found.

  1. Bennett-Levy, J.; Powell, G.E. The Subjective Memory Questionnaire (SMQ). An investigation into the self-reporting of ‘real-life’ memory skills. Br. J. Soc. Clin. Psychol. 1980, 19, 177–188, doi:10.1111/j.2044-8260.1980.tb00946.x.
